# Molecular Mechanisms of Lymphatic Metastasis in Breast Cancer: An Updated Review

**DOI:** 10.3390/cancers17132134

**Published:** 2025-06-25

**Authors:** Fatema Mahjabeen, Samrin F. Habbani, Sulma I. Mohammed

**Affiliations:** 1Department of Internal Medicine, Texas Tech University Health Sciences Center at Amarillo, Amarillo, TX 79106, USA; fatema.mahjabeen@ttuhsc.edu; 2Department of Comparative Pathobiology, Purdue University Institute of Cancer Research, Purdue University, West Lafayette, IN 47907, USA; shabbani@purdue.edu; 3Department of Small Animal Clinical Sciences and Cancer Control and Population Sciences, University of Florida Health Cancer Center, University of Florida, Gainesville, FL 32610, USA

**Keywords:** breast cancer metastasis, lymphatic metastasis, Lymph-Circulating Tumor Cells (LCTCs), molecular mechanisms, Epithelial-to-Mesenchymal Transition (EMT), Biomarkers

## Abstract

Breast cancer is the most common cancer among women and often results in poor outcomes when primary tumor cells spread beyond the breast to other parts of the body (metastasis). Tumor cells can spread through lymph nodes, particularly the sentinel lymph nodes, or through the bloodstream. Researchers are actively investigating circulating tumor cells, which are the known enemy, and the specific mechanisms through which cancer spreads from these nodes to other organs. This review explores the molecular and cellular mechanisms that enable breast cancer (circulating tumor cells) to metastasize while focusing on the lymphatic dissemination techniques of cancer cells, also highlighting emerging therapeutic strategies that target key lymphatic molecules and pathways for improving the prevention and treatment of breast cancer metastasis, ultimately enhancing patient outcomes.

## 1. Introduction

Breast cancer (BC) remains a significant health challenge globally and is the leading cause of cancer-related mortality among women worldwide [1,2,3]. Despite the substantial advancements in early detection and treatment, the disease accounted for 670,000 deaths worldwide in 2022 [1]. Patient prognosis is strongly influenced by the stage at diagnosis and the molecular subtype of the tumor [3]. Moreover, the presence of lymph node metastasis is one of the most important prognostic factors, and it is routinely used in staging and treatment decisions [4]. An estimated 20–30% of early-stage patients still develop metastatic disease within a decade, even after primary tumor excision [2,5]. Overall, metastasis is responsible for approximately 90% of breast cancer-related deaths [2].

Metastatic progression is governed by intricate molecular and cellular processes, most notably, uncontrolled proliferation, invasion, and migration, all of which are influenced by interactions with the tumor microenvironment (TME) [3,6]. These interactions drive immune system evasion, epithelial–mesenchymal plasticity (EMP), angiogenesis, and ultimately, tumor cell invasion into vascular structures (vascular invasion), particularly lymphatic capillaries, which play a key role in early dissemination [7,8].

Lymphatic vessel invasion (LVI) refers to the infiltration of tumor cells into endothelium-lined vascular structures, specifically lymphatic and blood vessels [9,10,11,12]. Notably, in breast cancer, immunohistochemistry studies have shown that the majority of LVI events in BC involve lymphatic rather than blood vessels [8].

Recent evidence suggests that lymphatic vascular biomarkers such as the Vascular Endothelial Growth Factor (VEGF)-R3, LYVE-1, podoplanin, and Prox-1 (prospero-related homeobox-1) play crucial roles in promoting lymphangiogenesis and facilitating tumor dissemination to axillary lymph nodes [13,14]. Furthermore, lymph circulating tumor cells (LCTCs) exhibit distinct biological properties compared to blood circulating tumor cells (BCTCs), showing increased metastatic potential and a unique transcriptomic profile that favors immune invasion and lymphatic dissemination [2,15].

Critical signaling pathways, governed by numerous genetic and somatic changes, can lead to aberrations in cellular differentiation, genomic stability, and growth. Many of these pathways, such as Notch, Wnt, NF-kB, Sonic Hedgehog (SHH), ER, PI3K/Akt/mTOR, and HER2, are emerging as promising therapeutic targets in breast cancer [16]. In this context, the axillary lymph nodes serve as both a primary conduit for metastatic breast cancer progression and a key prognostic indicator [17,18]. However, despite technological advancements, the molecular mechanisms governing lymphatic metastasis and systemic tumor dissemination remain incompletely understood. This gap in knowledge highlights a significant barrier to improving metastatic breast cancer management and underscores the urgent need for targeted therapeutic innovation [19].

This review focuses on the molecular mechanisms of lymphatic metastasis in breast cancer. We highlight the emerging therapeutic strategies based on recent advances in breast cancer metastasis research, with a particular emphasis on key molecular determinants, including lymphatic invasion, the tumor microenvironment (TME), and the lymphatic microenvironment (LME). Additionally, we examine the roles of immune-responsive cells, epithelial-to-mesenchymal transition (EMT), EMT-associated transcription factors, and cancer stem cells, in promoting tumor invasiveness and lymphatic dissemination. Moreover, we explore how epigenetic modifications contribute to breast cancer metastasis and offer targets for therapy. By integrating the current insights from diverse aspects of breast cancer metastasis biology, this review aims to advance our understanding of lymphatic spread in breast cancer, identify promising molecular pathways, and propose innovative therapeutic approaches. Ultimately, we hope to bridge critical knowledge gaps and contribute to improved clinical outcomes for patients with metastatic breast cancer.

## 2. Multivariate Breast Cancer Classification and Relation to Lymphatic Metastasis

Breast cancer is classified using a combination of histological, molecular, and receptor-based features to guide the association of clinical trials with the correct patient population and treatment regimens and therefore, develop therapies that are precise and targeted with regard to particular stages, grades, and subtypes. Histopathologically, breast tumors are categorized based on their origin in the ductal or lobular epithelium [3,20]. Invasive ductal carcinoma (IDC) accounts for about 60 to 75% of tumors, while invasive lobular carcinoma (ILC) constitutes about 10 to 15% of cases [3]. Although both subtypes can metastasize, ILC exhibits a distinct metastatic pattern, with a higher tendency to spread to the gastrointestinal tract and peritoneum [21], as well as includes more involvement of multiple lymph nodes compared to IDC [22].

At the molecular level, breast cancer is classified into four distinct subtypes—luminal A, luminal B, HER2-enriched, and basal or triple-negative breast cancer [3,23,24]. These molecular subtypes differ significantly in terms of tumor biology, therapeutic responses, prognosis, and metastatic behavior [24]. The luminal A subtype is associated with more favorable outcomes, whereas luminal B tumors are generally high-grade and more aggressive, characterized by elevated proliferation rates, as indicated by upregulation of Ki-67, a cell proliferative marker [3,25]. HER2-enriched tumors, defined by the overexpression of the HER2 oncogene, are associated with poor survival; however, anti-HER2 therapy has improved outcomes [23,24]. The most hostile tumor is the triple-negative subtype, which also has a poor survival chance and early metastasis spread, particularly to the lung, liver, bone, and brain [3]. Overall data about the clear-cut relation between breast cancer subtypes and the frequency of axillary lymph node involvement are mixed. Some studies showed a relatively low risk of axillary lymph node involvement in the basal molecular type of breast cancer [26]. Certain studies indicated that triple-positive patients are more likely to have nodal metastases, while others reported that there was no significant difference in lymph node involvement in ER+/PR+/HER2+ patients. Conversely, some researchers suggest that hormone receptor-positive, HER2-enriched subgroups may exhibit the lowest rates of node metastasis [26].

## 3. Evolution of Breast Cancer Treatment in the Context of Lymphatic Dissection

Breast cancer treatment strategies have undergone significant evolution, reflecting advances in our understanding of tumor biology and metastasis. By carefully examining the various timelines of treatment protocol transformations in breast cancer, we gain insight into the lymphatic system in terms of disease progression and why the continuous demand requires more attention and intervention.

### 3.1. From Radical Surgery to Systemic Therapy

Initially, breast cancer management focused on controlling localized disease through surgery and radiation. Advancements in molecular biology have introduced systemic therapies and personalized approaches, like targeted therapy that targets both primary tumors and metastatic disease [2]. The radical mastectomy, derived from Halsted’s theory that breast cancer mainly spreads through the axillary lymph nodes, became the standard procedure. This involved removing the primary breast tumor, the underlying pectoralis muscle, and the axillary lymph nodes, aiming for a potential cure.

Later, the systemic nature of breast cancer was proposed by researchers from the National Surgical Adjuvant Breast and Bowel Project (NSABP), who suggested that breast cancer cells could enter the bloodstream directly from the primary tumor site [27]. This indicates that metastasis to regional lymph nodes, such as axillary lymph nodes, might occur randomly and not necessarily in all breast cancer patients. This insight shifted treatment towards a systemic approach [15,28,29].

Subsequently, the addition of radiation and chemotherapy as both neoadjuvant (pre-surgery) and adjuvant (post-surgery) therapies has further improved treatment outcomes and expanded management options [18,30]. Importantly, research indicates that the lymphatic system plays a crucial role. Lymphatic vessel invasion has been identified as a strong predictor of both lymph node and distant metastasis [4,8]. Pathologists, using immunohistochemistry and lymphatic vessel biomarkers, have found that over 95% of the vessels involved are lymphatic in origin [8]. A study conducted by Nathanson et al. explored the relationships between regional lymph nodes (RLN) and systemic metastasis, revealing that lymphatic metastasis is a significant and independent predictor of both RLN and systemic metastasis, especially when RLN metastasis is present [31]. These results support that lymphatic spread is not just a local phenomenon but can also seed distant metastasis.

### 3.2. Advances in Surgical Approaches

To reduce the morbidity associated with complete axillary lymph node dissection (CALND), particularly lymphedema, sentinel lymph node biopsy (SLNB) emerged as a less invasive alternative. After extensive validation, SLNB was established as a standard of care for accurately detecting lymph node metastasis [32,33]. In clinical practice, lymphatic tracers such as radionuclide-labeled sulfur colloids, blue dye, and indocyanine green are commonly used to identify the location of SLNs [33].

Moreover, advances in systemic therapies, such as anti-HER2 treatment, have further reduced the need for CALND. For example, 74% of lymph node-positive patients treated with neoadjuvant therapy became node-negative [32,34]. Consequently, breast-conserving surgery is now widely favored due to its better patient outcomes and reduced morbidity compared to CALND [35,36]. SLN biopsy, preoperative chemotherapy, and breast-conserving surgery are standard approaches for effective disease control in both node-negative and node-positive patients [35].

### 3.3. Experimental Models and the Emerging Role of Lymph Nodes in Metastasis

Recent studies in in vivo models have provided important insights into the metastatic dissemination of primary breast tumors through both lymphatic and vascular systems [2,15,28,29]. Techniques such as fluorescent protein tagging and direct inoculation of cancer cells into afferent lymphatic channels have been employed to track these changes [15,28,29]. Although this is not yet standard practice, there is increasing evidence suggesting that lymph nodes may be potential sources of metastases.

For example, Pereira et al. employed the photo-adaptable fluorescent protein Dendra2 to trace metastatic cancer cells traveling from primary tumors to lymph nodes in mice with mammary tumors. They orthotopically implanted pDendra2-tagged triple-negative breast cancer cells (4T1, ATCC, Manassass, VA, USA), melanoma cells (B16F10, ATCC, Manassas, VA, USA), and squamous cell carcinoma cells (SCCVII, Frederick National Laboratory, Frederick, MD, USA) into syngeneic mouse models. After excising the primary tumor and selectively exposing the lymph nodes to a 405 nm laser diode, metastatic cells in the lymph nodes photoconverted to emit red fluorescence, while primary tumor cells retained green fluorescence [29]. Despite its utility, this system has limitations, including incomplete photoconversion efficiency (~70%) and fluorescence reversion in metastatic lymph node cells during cell division, making it challenging to accurately determine the origins of distant metastases [8,26].

Further investigations using intravital staining microscopy and molecular markers, such as anti-cytokeratin (cancer cells), anti-CD31 (blood vessels), and collagen IV (vascular basement membrane), revealed that cancer cells preferentially interacted with the basement membranes of blood vessels. These observations support the hypothesis that distant metastasis in breast cancer may originate not only from the primary tumor but also from sentinel lymph nodes (SLN), regional lymph nodes (RLNs), or axillary lymph nodes (ALNs) [37]. In complementary studies, Brown et al. conducted a micro-inoculation of human mammary cancer cells into the afferent lymphatic channels of mice. They observed the cells infiltrate the lymph node parenchyma and blood vessels before seeding in the lungs [37]. These findings further support the concept that lymph node vasculature may serve as exit routes for systemic dissemination [31,37].

## 4. Mechanisms of Lymphatic Invasion and Migration

Tumor cell migration is a crucial initial step in breast cancer progression and metastatic spread. While breast cancer can disseminate through both the blood and lymphatic routes, increasing evidence suggests the lymphatic system’s critical role in early-stage metastasis [4,15,38,39,40]. Lymphatic spread begins with malignant epithelial cells breaching the basal lamina of the breast duct and transitioning from being an in situ disease to invasive carcinoma. These tumor cells then penetrate the lymphatic vessels, travel to lymph nodes, and may eventually seed distant organs. This process is regulated by a network of signaling pathways, growth factors, immune modulators, and cellular adaptations [24,38,39]. Figure 1 provides a schematic overview of the key steps involved.

### 4.1. Lymphangiogenesis and Immune Modulation

The first major event in lymphatic metastasis is tumor-induced lymphangiogenesis—the formation of new lymphatic vessels. The process is initiated by the tumor microenvironment, which secretes lymphangiogenic growth factors, including VEGF-C and VEGF-D. These factors bind to the specific receptor VEGF-R3 on LECs [33,39,40,41]. VEGF-R3 activation promotes LEC proliferation, leading to the formation of new lymphatic vessels around the tumor, increased lymphatic endothelial permeability, and the enlargement and remodeling of pre-existing lymphatics [40,42,43,44]. This expansion of peritumoral lymphatic vessels facilitates the entry of tumor cells into the lymphatic circulation (Figure 1).

Several studies have emphasized the role of other specific lymphatic markers, including lymphatic vessel endothelial hyaluronic acid receptor 1 (LYVE-1), prospero homeobox 1 (Prox1), and podoplanin, as key regulators that facilitate a clear distinction between lymphatic and blood vessel invasion by breast cancer tumor cells [39,40]. Using these markers in conjunction with double immunostaining yielded extensive evidence for lymphangiogenesis [40].

Additionally, the immune modulation of the lymphatic niche plays a crucial role in regulating lymphangiogenesis. Immune cells, such as tumor-associated macrophages (TAMs), significantly contribute to lymphangiogenesis by producing VEGF-C, which binds to VEGFR-3, promoting endothelial cell proliferation and tumor-induced lymphangiogenesis via PI3K/Akt and MEK/ERK signaling pathways [45]. Correlation analyses have shown a direct association between TAM abundance, VEGF-C expression, and lymphatic microvessel density (LMVD), thereby linking TAMs to lymphatic metastasis [46]. While TAMs have multifaceted origins and functions, one subset identified in murine models of breast tumors is LYVE-1 receptor-expressing TAMs. These LYVE-1-positive TAMs are capable of forming a specialized multicellular niche along the peritumor vascular territory, which is activated by IL-6 and, in turn, activates CCR5-dependent signals, including heme oxygenase-1. This particular subset of TAMS can be a crucial target for therapy, as genetic targeting in a mouse model has demonstrated improved chemotherapy response with enhanced CD8 T cell function within tumors [47]. TAMs also produce CXCL12, which binds to CXCR4 receptors on tumor cells, guiding their intravasation into lymphatic channels. Conversely, TAMs (specifically M2 macrophages) support tumor invasion by secreting mediators like MMPs and epidermal growth factor (EGF), promoting lymphatic intravasation and distant metastasis [48,49]. Cancer cells secrete colony-stimulating factors (CSFs), which attract M2 macrophages to tumors in a reciprocal relationship [50]. Additionally, primary tumors may induce macrophages to release indoleamine-pyrrole 2,3-dioxygenase (IDO), impairing T cell anti-tumor activity and facilitating immunological tolerance [51]. The CD163+ macrophages stimulate the CXCR4/CXCL12 axis in primary tumors and SLNs, further driving the migration and metastasis of luminal B BC cells [52].

Cancer-associated fibroblasts (CAFs) also modulate the lymphatic niche. These cells express podoplanin (D2-40) and activate the Wnt/beta-catenin pathway, increasing MMP secretion and facilitating vessel wall degradation and tumor cell intravasation [33,53,54,55]. Notably, express podoplanin (D2-40) expression in CAFs is linked to poor prognostic features like nodal invasion and poor differentiation [33].

Furthermore, pro-inflammatory cytokines such as IL-17A and IL-1β facilitate breast cancer cell adhesion to the lymphatic endothelium and subsequent transmigration [56,57]. Adhesion molecules, including N-cadherin and VE-cadherin, also influence tumor progression. N-cadherin suppresses E-cadherin transcription, while VE-cadherin promotes epithelial-to-mesenchymal transition (EMT) and enhances tumor invasiveness through the transforming growth factor-beta (TGF-β) pathway [58,59]. Stress-induced proteins, such as vimentin, contribute to EMT, increasing tumor cell motility and adhesion strength through activation of the Rac pathway, as observed in MCF-7 cells [15,44,60,61].

### 4.2. Tumor Cell Migration

Breast tumor cells detach from the extracellular matrix (ECM) and initiate local invasion and migration. Breast cancer can migrate as a single cell or in a coherent cluster. In single-cell migration, cancer cells utilize a protease-dependent pathway involving the proteolytic cleavage of the mesenchyme by enzymes such as matrix metalloproteinases (MMPs) or a protease-independent ameboid movement driven by cytoskeletal forces, allowing rapid navigation through the ECM without degrading it [42,62,63]. In collective migration, cancer cells move in clusters, guided by actin filament polymerization and lamellipodial protrusions. These cellular protrusions form focal adhesions with ECM components like fibronectin via beta-1-integrin, while contractile forces from the actin–myosin complex propel the clusters forward. Essential regulatory proteins such as talin, cortactin, paxillin, and vinculin contribute to the spatial and temporal coordination of this process [64,65,66,67,68].

In the context of lymphatic invasion, tumor cells are particularly adapted to the unique features of lymphatic vessels, such as a thin wall. Chemokine signaling, particularly through CCR7 and its ligand, CCL21, expressed by lymphatic endothelial cells, also directs tumor cells toward the lymphatic system [43,44,69].

### 4.3. Molecular Drivers of Lymphatic Invasion

To invade lymphatic vessels, breast cancer cells form invadopodia—specialized actin-rich protrusions that degrade the ECM and the basement membrane. These structures secrete MMPs and other proteolytic enzymes to facilitate intravasation [33,44,63,70].

MMPs further contribute to metastasis by degrading the ECM and producing chemoattractants, such as the hepatocyte growth factor (HGF), which reorganizes actin filaments through the Rho and Ras pathways, thereby sustaining membrane protrusions and directional migration [71,72,73].

Several molecular mediators promote lymphatic invasion and metastasis by regulating inflammation, lipid metabolism, and vascular remodeling. Among them, NF-κB signaling plays a role in lymphangiogenesis by driving the transcription of multiple inflammatory genes, including TNF-α, IL-1β, IL-6, IL-8, and IL-7, as well as enhancing the expression of VEGF-C [38,40,44]. Cyclooxygenase-2 (COX2) has been shown to drive lymphangiogenesis and lymph node metastasis in breast cancer by upregulating VEGF-C expression, enhancing LEC activation, and remodeling. It also participates in the prostaglandin E2 (PGE2)/EP signaling cascade, which influences dendritic cell function and promotes the formation of a pre-metastatic niche in regional lymph nodes [43,44].

### 4.4. Transcriptional and Metabolic Adaptation for Lymphatic Spread

Emerging evidence from single-cell and spatial transcriptome studies has revealed distinct mechanisms underlying lymphatic metastasis in breast cancer. One of the recent advancements to provide new insights into genetic programming or specific identification of markers and regulators for lymph node metastasis is various types of “-omic” studies, including but not limited to genomic, transcriptomic, proteomic, and epigenomic studies. These studies are better equipped to identify alterations in gene expression resulting in different protein biomarkers when comparing primary versus matched lymph node disseminated breast cancers in patients. They can also contribute to the prediction of clinical benefits from various therapeutic options, particularly for lymph node positive patients [74,75,76,77,78,79,80]. For example, single-cell RNA sequencing (scRNA-seq) analyses have shown that lymphatic circulating breast cancer cells often exhibit unique transcriptional profiles. Xu et al. conducted a study profiling 27,028 single cells from primary cancer tissue and 69,768 single cells from matched axillary lymph nodes, identifying a subpopulation of cancer cells with elevated oxidative phosphorylation (OXPHOS) activity, suggesting a metabolic change that may facilitate lymphatic spread [81]. Additionally, these lymph-invading cells have upregulated genes associated with immune evasion, such as NECTIN2. NECTIN2 interacts with TIGIT on immune cells, potentially supporting immune escape during lymph node colonization [81,82].

## 5. Lymphatic Microenvironment

The lymphatic system, particularly sentinel lymph nodes, is a common early site for the dissemination of breast tumor cells [44,61,83]. In addition to their role in metastasis, the lymphatic system is essential for transporting interstitial fluid, lipids, and immune cells from peripheral tissues to lymph nodes. This transportation maintains fluid balance and supports immune responses, which tumors can exploit to evade immune detection [61].

Unlike the bloodstream, the lymphatic system offers a permissive niche for tumor cells to survive. Recent studies suggest that the lymphatic system actively influences the behavior of tumor cells [15,84,85]. LCTCs dynamically interact with immune cells, stromal elements, and cytokines, which can trigger EMT, enhance stem-like traits, and promote cell clustering [15,84,85]. Mohammed et al. (2019) emphasized that the lymphatic system is not merely a pathway for tumor dissemination but also a specialized environment that shapes LCTCs [15]. These interactions not only support tumor survival and immune evasion but also promote epithelial plasticity and metastasis potential. Importantly, LCTCs exhibit characteristics of cancer stem cells (CSCs), expressing markers such as CD24, ALDH1 (epithelial), and CD44 (mesenchymal), indicative of a hybrid epithelial/mesenchymal phenotype [86,87]. Figure 2 illustrates the sequential breast cancer metastasis through the lymphatic system. It depicts tumor cell migration, intravasation into lymphatic vessels, dissemination as LCTCs, and extravasation into lymph nodes or distant metastatic sites. At the center of this metastatic cascade are LCTCs, characterized by a hybrid epithelial–mesenchymal phenotype (e.g., CD44+/CD24+), which enhances their ability to cluster, evade immune surveillance, and colonize lymphatic tissues.

## 6. Epithelial-to-Mesenchymal Transition (EMT) in Lymphatic Dissemination

Epithelial-to-mesenchymal transition (EMT) is a critical biological process that enables epithelial tumor cells to acquire mesenchymal traits, facilitating invasion, migration, and intravasation into the lymphatic system. It is a process in which epithelial cells acquire mesenchymal characteristics, resulting in the loss of intercellular adhesion, increased mobility, and enhanced invasive properties (Figure 3) [42,88,89,90,91].

EMT occurs reciprocally during metastasis: epithelial cells undergo transdifferentiation into mesenchymal cells to escape the primary tumor, invade lymphatic vessels, and migrate to distant sites. Once at the secondary organ, these cells undergo a mesenchymal-to-epithelial transition (MET), re-differentiate into their epithelial origins, and restore tight intercellular connections (Figure 3) [92,93]. This dynamic process is regulated by transcription factors, signaling molecules, and surface proteins. Key EMT transcription factors—such as Snail, Slug, and Twist—play pivotal roles in initiating the EMT by repressing E-cadherin expression and upregulating mesenchymal genes. TWIST1, for instance, recruits chromatin remodeling complexes that epigenetically silence E-cadherin, promoting cell motility and stem-like traits. Furthermore, lymph vessels can act as a more conducive route due to their thin walls and discontinuous basement membranes. There is an upregulation of certain chemokine receptors, such as CCR7, which directs cells towards lymphatics expressing CCL21, the corresponding ligand [91].

Breast cancer cells can undergo a phenotypic shift during their EMT process through the upregulation of N-cadherin and vimentin, and the downregulation of E-cadherin, which reduces intercellular adhesion. These mesenchymal features help breast cancer cells to invade lymphatic vessels [94]. Markiewicz et al. in their study showed that immunohistochemical and quantitative reverse transcriptase PCR from metastatic lymph nodes and primary tumors of patients with breast cancer reveal increased mRNA expression and protein levels of EMT regulators (TWIST1, SNAIL), which are associated with lower overall survival and disease-free survival, thereby signifying a more aggressive phenotype of cancers [87].

EMT-inducing transcription factors, such as TWIST1 and SNAIL1, not only repress E-cadherin but also drive tumor angiogenesis through the regulation of MMP9, N-cadherin, and vimentin. For example, TWIST1 engages the TWIST1/Mi2/NuRD complex, which epigenetically silences E-cadherin expression via promoter hypermethylation [95,96,97]. Inflammatory cytokines, such as interleukin-6 (IL-6), further exacerbate metastasis by activating the JAK/STAT3 signaling pathway, linking inflammation to tumor proliferation and dissemination [98].

Moreover, EMT is closely linked to cancer stem cell (CSC) properties [86,99,100]. CSCs possess unique properties, including self-renewal, differentiation, tumorigenicity, and resistance to therapy [101]. Lin28, particularly Lin28B, is a critical regulator of BCSCs, promoting metastasis in experimental models [102].

Lin28B, an RNA-binding protein, contributes to breast cancer metastasis through multiple mechanisms, including the enhancement of cancer stem cell properties and the formation of a pre-metastatic niche through immunosuppression. Upregulation of this in breast cancer cells leads to the secretion of exosomes deficient in let-7 microRNAs, facilitating the recruitment of N2 phenotype neutrophils. N2 neutrophils secrete cytokines such as IL-6 and IL-10 and upregulate PD-L2, thereby suppressing T cell function and creating an environment conducive to colonization [103]. Lin28B is also associated with the overexpression of breast cancer stem cells (CSCs), along with ALDH1 and OCT4, which increases tumorigenicity and invasiveness. Moreover, dysregulated pathways, such as the Wnt/β-catenin, Hedgehog, Notch, and PI3K/Akt/NF-κB pathways, drive BCSC proliferation, EMT activation, and therapeutic resistance [104]. For example, the Wnt/β-Catenin Pathway promotes BCSC stemness and EMT. Abnormal activation leads to the transcription of genes that enhance proliferation and inhibit differentiation, resulting in the accumulation of β-catenin in the nucleus. Inhibition of this pathway with LGK974 and niclosamide can potentially reduce BCSC populations and tumorigenicity. Similarly, the use of inhibitors targeting components such as Smoothened (SMO) has shown promise in disrupting Hedgehog (Hh) signaling in BCSCs, thereby reducing BCSC self-renewal and survival. The PI3K/Akt/NF-κB pathway is associated with chemoresistance, and therefore, inhibitors of PI3K or Akt can sensitize BCSCs to chemotherapy [105].

Certain differences between LCTCs and BCTCs help characterize molecular events in breast cancer metastasis. LCTCs, even when they undergo EMT, possess both the properties of epithelial and mesenchymal cells and follow different TGF-b and EMT paths. LCTCs, in comparison to BCTCs, can stay in clusters while metastasizing, and cluster cells have been known to be more efficient in metastasis than cells moving discretely [15,106].

This co-expression of both epithelial and mesenchymal markers enables LCTCs to adopt a hybrid epithelial/mesenchymal (E/M) phenotype, which has been linked to making them more efficient for mammosphere formation and stemness [107]. Additionally, LCTCs have been shown to co-express E-cadherin, vimentin, and N-cadherin, further confirming the hybrid E/M state [15]. Notably, few studies, including Grosse-Wilde et al. (2015), have reported that the hybrid E/M state is associated with enhanced stemness in breast cancer cells and may correlate with poor patient survival and unfavorable clinical outcomes [107].

## 7. The Tumor Microenvironment in Metastatic Breast Cancer

The tumor microenvironment (TME) is a dynamic network of stromal cells, immune cells, and ECM components that collectively drive tumor progression and metastasis. In breast cancer, the TME consists of a diverse mixture of immune and non-immune cells of mesodermal origin, along with the proteins they secrete and the ECM in which they reside. The TME plays a pivotal role in carcinogenesis by maintaining a balance between tumor-suppressing cytotoxic cells and tumor-supporting cells like M2 macrophages and myeloid-derived suppressor cells (MDSCs), which are recruited by inflammatory cytokines such as CCL2 [108,109].

### 7.1. Cellular and Molecular Composition

The interaction between the TME and tumor cells closely resembles the influence of the embryonic mesenchyme on epithelial differentiation. While the normal breast microenvironment regulates mammary ductogenesis, the tumor microenvironment facilitates epigenetic changes in stromal cells, promoting carcinogenesis [6,110]. Non-immune cells within the TME, including cancer-associated fibroblasts (CAFs), adipocytes, and endothelial cells, indirectly support tumor progression by providing cytokines and signaling molecules that enhance angiogenesis, immune evasion, and vascular invasion. CAFs, for example, stimulate angiogenesis by secreting the stromal-derived factor (SDF)-1/CXCL12, which interacts with CXCR4 in a paracrine manner [110,111].

### 7.2. Gene Expression and Signaling

Gene expression changes in TME components also contribute to tumor growth. In ductal carcinoma in situ (DCIS)-associated myoepithelial cells, genes regulating laminin and oxytocin receptor function are downregulated, while tumor-promoting genes like CXCL12 and CXCL14 are upregulated. These changes facilitate the transition from DCIS to invasive ductal carcinoma (IDC). Experimental models have shown that downregulating differentiation regulators such as TGFBR2, SMAD4, or GLI1 in myoepithelial cells accelerates tumor invasion [112]. Additionally, CAFs increase IL-6 and IL-8 secretion in tumor cells, promoting invasion and angiogenesis [113].

Immune cells in the TME, including macrophages, neutrophils, and dendritic cells, are recruited by tumor-released chemoattractants and cytokines such as VEGF, tumor necrosis factor-alpha (TNF-α), and angiopoietin-2. Tumor-associated macrophages (TAMs), in particular, become pro-angiogenic, producing vascular-modulating enzymes like MMP9 and growth factors such as VEGF, CCL2, and CXCL8. Prolonged stimulation can even induce TAMs to transform into vascular endothelial cells. Neutralizing antibodies targeting VEGF or CCL2 have shown promise in reducing tumor angiogenesis and progression [114].

### 7.3. Targeting the TME

The Notch signaling pathway plays a central role in the TME, regulating tumor cell proliferation, survival, migration, and intravasation [115,116]. This is one of the highly conserved cell-to-cell communication mechanisms that is known to regulate cell fate decisions. Notch ligands (e.g., Jagged1) on a cell bind to Notch receptors on an adjacent signal-receiving cell, activating the signal and triggering two subsequent proteolytic cleavages of the Notch receptor: first by ADAM-family metalloproteases and followed by γ-secretase. The resultant protein cleavages release the Notch intracellular domain (NICD), which then translocates to the nucleus and associates with DNA-binding proteins to regulate the transcription of target genes involved in cell differentiation, proliferation, and apoptosis [117]. Notch signaling can exhibit dual roles in the epithelial-to-mesenchymal transition (EMT), acting as both a promoter and an inhibitor, depending on the cellular context and the specific Notch receptors involved [118]. Once activated, Notch1 induces EMT by upregulating transcriptional repressors SLUG and SNAIL, leading to the downregulation of E-cadherin and promoting a mesenchymal phenotype. Notch3, on the other hand, has been reported to inhibit EMT in breast cancer cells by activating the Hippo/YAP pathway through the upregulation of Kibra, thereby maintaining epithelial characteristics and suppressing metastasis [118].

It also interacts with VEGF signaling to modulate angiogenesis and remodel the ECM by inducing NF-κB, MMP2, and MMP9 expression [117,119,120,121]. Increased urokinase plasminogen activator (uPA) production driven by Notch signaling promotes ECM degradation, enhancing metastasis [122]. Inhibiting Notch signaling using γ-secretase inhibitors (GSIs) such as PF-03084014 has shown significant efficacy in preclinical models, particularly in triple-negative breast cancer (TNBC). GSIs induce tumor burden reduction and cell cycle arrest, providing a potential therapeutic avenue for metastatic breast cancer [123,124]. Additionally, treatments such as BXL0124, a Gemini vitamin D derivative, have demonstrated efficacy in reducing the cancer stem cell population by blocking Notch1 signaling [125].

## 8. The Role of Epigenetic Regulation in Lymphatic Metastasis of Breast Cancer

Epigenetic changes, including DNA methylation, histone modifications, and non-coding RNAs, play a crucial role in regulating the expression of key genes implicated in metastatic breast cancer. Although epigenetic mechanisms are part of normal development, disruptions in these processes can modify gene expression via DNA methylation, histone modifications, microRNAs, and nucleosome remodeling. These disruptions convert normal cells into malignant ones with high metastatic potential [24,126,127]. Importantly, epigenetic transformations are reversible, stable, and inheritable, offering promising avenues for cancer intervention [128,129].

Several studies have also examined the role of epigenetic alterations in driving the lymphatic metastasis of breast cancer. Some studies are focused on the effective DNA methylation of subset of cancer genes like DFNA 5, ID4, CDH 1, which is found to have a positive correlation with lymph node positive breast cancer patients [26,130,131,132,133,134]. Because of the heterogeneous findings between primary breast tumors and matched lymph node metastasis patients nowadays, genome-wide DNA methylation analysis was performed in the cohort of patients with triple-negative breast cancers matching with the counterpart of normal adjacent and lymph node metastasis tissues [135]. This unique study showed almost 83 genes had altered expression in the primary tumor versus lymph node metastasis cases. 18 genes among those 83 lymph node-associated genes were found to have altered DNA methylation patterns [26,135].

Further workup by Leslie et al. on a syngeneic mouse model in an attempt to show the effect of histone deacetylase 11 (HDAC11) on lymph node dissemination revealed that the inhibition of HDAC 11 might decrease intranodal tumor cell proliferation but paradoxically substantially increases migration of breast cancer cells from the lymph node to the lung [136]. The above-mentioned studies did not adequately shed light on the functional validation of the effect of differential genetic expression and methylation patterns on lymph node metastasis. This barrier of tumor heterogeneity can be overcome by single-cell sequencing technologies. Bow et al. utilized laser-capture microdissection techniques to isolate tissue from a primary tumor, adjacent morphologically normal ductal tissue and sentinel lymph node, then used single-cell sequencing to identify the clonal expansion of a single tumor cell which had overexpression of the pro survival gene, MCL 1 and ch8q as potential drivers of lymphatic metastasis of tumor cells [137].

## 9. Conclusions and Future Perspectives

Breast cancer, being a highly heterogeneous disease, progresses to metastasis through diverse pathways, including lymphatic and hematogenous routes. This review explored advancements in understanding the mechanisms and routes of breast cancer metastasis. One limitation of this review is that it focuses on a single database search where we tried to keep the focus on lymphatic metastasis mechanisms. This review did not go in depth about the metabolic aspects of cancers cells, especially how these might differ at genetic levels when undergoing lymphogenous versus hematogenous metastasis. Nevertheless, this review unravels insights that are essential for clinicians, oncologists, and researchers to develop innovative therapies and optimize existing treatment strategies. Given the substantial global burden of breast cancer metastasis, identifying preferential metastatic pathways offers an opportunity to target key molecules, genes, or biomarkers for therapeutic intervention.

Despite extensive research, no single pathway or molecule has emerged as a universal therapeutic target suitable for the broader population. The complexity of the metastatic process underscores the need for continued investigations into the genes, proteins, and signaling cascades involved in lymphatic invasion and metastatic spread. Unraveling these molecular mechanisms will pave the way for novel therapeutic approaches, ultimately improving disease-free survival rates and the quality of life for breast cancer patients.

Future efforts should focus on designing customized treatment plans tailored to the unique metastatic potential of individual tumors. This precision-based approach would target the specific mechanisms underlying cancer cell dissemination and migration, offering a promising pathway toward more effective and personalized care for breast cancer patients.

## Figures and Tables

**Figure 1 cancers-17-02134-f001:**
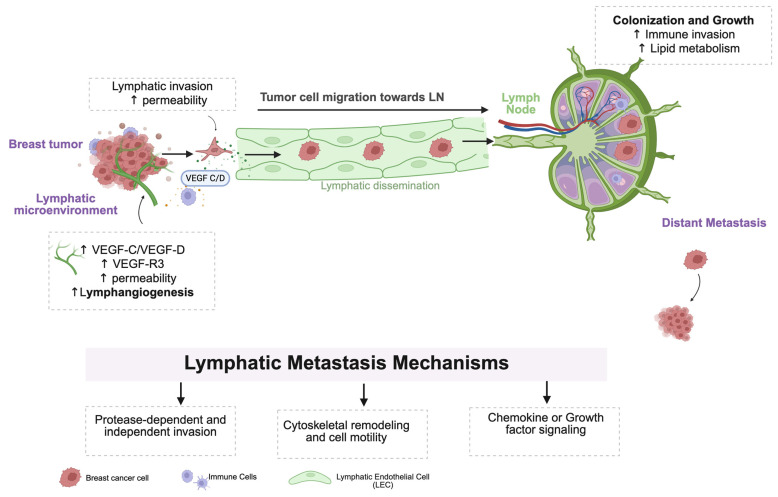
Schematic representation of key stages in lymphatic metastasis: Cancer cells undergo lymphangiogenesis-driven entry into lymphatic vessels, followed by cytoskeletal remodeling, and attain cell motility. Subsequent steps include protease-dependent and -independent invasion, signal integration in directed migration, focal adhesion dynamics, and chemokine or growth signaling. Arrows indicate the direction of tumor cell migration and signaling events involved in metastatic progression towards lymph nodes and distant sites.

**Figure 2 cancers-17-02134-f002:**
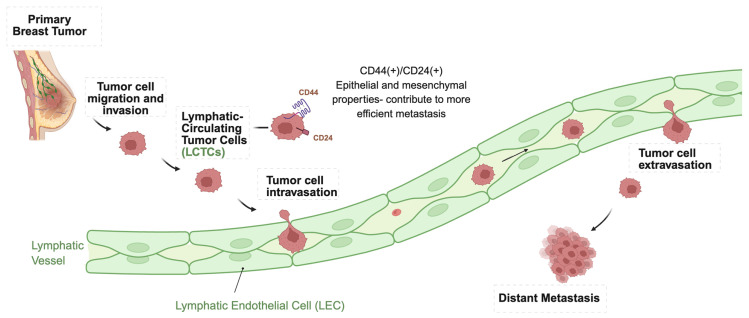
Schematic illustration of breast cancer metastasis via lymphatic circulating tumor cells (LCTCs). Tumor cells originating from the primary tumor site acquire invasive potential and enter the lymphatic system as lymphatic circulating tumor cells (LCTCs). These cells, characterized by a hybrid epithelial–mesenchymal phenotype (CD44+/CD24+), disseminate through lymphatic vessels, undergo intravasation and extravasation, and ultimately colonize distant metastatic sites. Arrows show the direction of tumor cell migration and invasion through the lymphatic system, leading to distant metastases.

**Figure 3 cancers-17-02134-f003:**
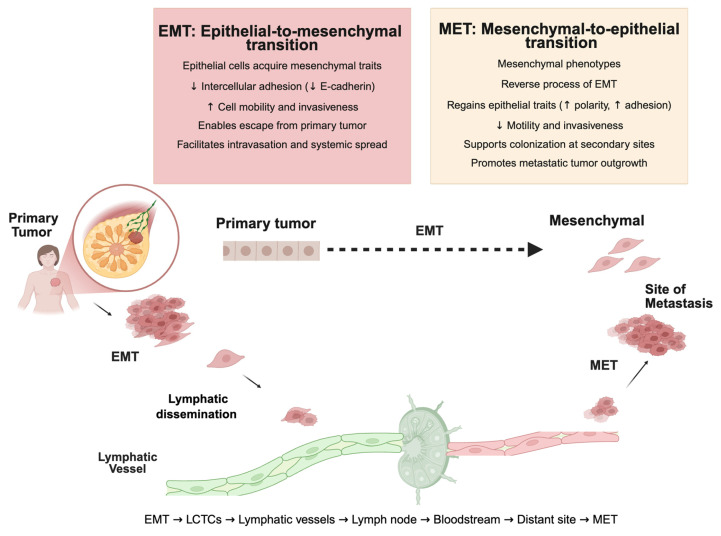
Epithelial tumor cells at the primary breast tumor site undergo epithelial-to-mesenchymal transition (EMT), characterized by reduced intercellular adhesion and increased motility, facilitating escape into lymphatic vessels as lymphatic circulating tumor cells (LCTCs). These cells travel through lymphatic channels to the sentinel lymph node, where they may transit into the bloodstream and disseminate to distant organs. At metastatic sites, tumor cells may undergo mesenchymal-to-epithelial transition (MET), regaining epithelial characteristics that support colonization and outgrowth. The schematic highlights the sequential process: EMT → LCTCs → lymphatic vessels → lymph node → bloodstream → distant site → MET. Arrows in the figure indicate the directional flow of tumor cell migration.

## Data Availability

Not applicable; no new data was created or analyzed in this review.

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
