# Peer review of "Molecular Mechanisms of Lymphatic Metastasis in Breast Cancer: An Updated Review"

_cancers, 2025, doi:10.3390/cancers17132134_

Round 1
Reviewer 1 Report
Comments and Suggestions for Authors
The authors focus and cancers from 'the breast ducts.' This ignores the 10-15% of breast cancers arising from glands (lobes).
LVI is used repeatedly and usually in a relevant fashion. However, I would suggest using 'lymphatic' and 'blood vessel capillary' invasion specifically to indicate where breast cancer cells invade vessels. LVI is a vague term used by pathologists when they cannot distinguish lymphatic from blood vessel invasion. In a review of this nature, it is preferable to be specific.
Reviewer 2 Report
Comments and Suggestions for Authors
In this article, the authors summarized the recent progress on the molecular mechanism of lymphatic metastasis in breast cancer. The article is basically well written and interesting. The reviewer has only a few comments.
Comments.
1, Fig.2. We see “Lymphatic route” at the left, and “Primary tumor site” at the center. Is it O.K. for the relationship between these two words?
2, Fig.3: In the central area, we see two tissues, one with “(LCTCs)”. Does one of these indicate other breast tissues (such as intra-tissue metastasis) and the other a lymph node? Please indicate them more clearly.
Reviewer 3 Report
Comments and Suggestions for Authors
The authors provide a comprehensive review of lymphatic metastasis for breast cancer. They discuss the contribution of the tumor microenvironment and the molecular mechanisms. The review, however, covers metastasis generally without distinguishing the mechanism of lymphatic metastasis sufficiently. The authors could also provide a diagram of the various lymphatic metastasis mechanisms. Breast cancer cells have preferential homing sites for metastasis, such as brain or bone. How does lymphatic metastasis play into this? How do molecular mechanisms vary between blood circulation and the lymphatic system? The authors touch on various aspects of breast cancer, such as emt, and immune cells, but could elaborate on the role of these in steering cancer cells to intravasation vs lymphatic drainage. The authors could discuss the differences between cancer cells from lymph nodes and circulating tumor cells, which go directly to blood circulation. Are there omics types of studies delineating their differences, particularly newer single-cell technology studies, that could be a helpful addition to the review? Additionally, are there any subtype-specific/genetic ancestry-specific differences?
Reviewer 4 Report
Comments and Suggestions for Authors
This review summarizes the latest advances in molecular mechanisms related to lymphatic metastasis in the field of breast cancer. However, several issues need correction.
Line 64: Please add a period here.
Line 96: Since this is an updated review, what time range did your literature search focus on? Did you only use PubMed? Relying on a single database may be insufficient.
Line 101: Given that this review is not a meta-analysis, I believe the methodology section is unnecessary. However, there is no major impact if you wish to retain it.
Line 164: I suggest summarizing existing breast cancer tumor experimental models and dissemination mechanisms first, rather than starting this paragraph by introducing a single study.
Line 215: Which migration pathway of tumor cells occurs more frequently?
Line 232: Please add a period here.
Line 330: Please elaborate on the molecular regulation of the Notch signaling pathway—how does it conduct signal transduction, and how does it inhibit the EMT pathway?
Line 334: Please complete this sentence.
Line 354: How do inflammatory processes promote cancer cell proliferation? Please clarify this information.
Line 449: How does Lin28 promote breast cancer metastasis? Can you elaborate on its molecular mechanism?
Line 451: I believe it is also crucial to understand how these signaling pathways affect the BCSC process. Understanding this is essential for determining how to intervene in BCSC and cancer progression by regulating signaling pathways.
Conclusion: Include the shortcomings of this review.
Conclusion: Clarify the value of this review and its contributions to cancer research and medicine.
Round 2
Reviewer 4 Report
Comments and Suggestions for Authors
The revised manuscript is fine